# Impact of joint commission international accreditation on occupational health and patient safety: A systematic review

Anni Vuohijoki [1,2]*, Leena Ristolainen[2], Juhana Leppilahti[1], Sanna-Maria Kivivuori[3], Heikki Hurri[4]

1 Translational Medicine Research Unit, Medical Research Center Oulu, Oulu University Hospital, Oulu, Finland, 2 Orton Orthopaedic Hospital, Helsinki, Finland, 3 Helsinki University Hospital, Helsinki, Finland, 4 Research Institute Orton, Helsinki, Finland

* anni.vuohijoki@student.oulu.fi

## Abstract

### Objectives

To illuminate the benefits of Joint Commission International (JCI) accreditation and other experiences related to the accreditation process in relation to occupational health and patient safety.

### Design

Systematic review

### Methods

We systematically searched CINAHL (n = 77), ProQuest (n = 69), PsycINFO (n = 13), PubMed (n = 166), and Scopus (n = 211) for articles on JCI accreditation published until December 2023. Overall, 290 articles were found. JCI-accredited hospitals with before–after accreditation follow-up processes, hospitals during the accreditation process, and job satisfaction and/or patient safety as the primary outcome measures were included. Non-original articles, articles including non-accredited hospitals, hospitals missing before–after accreditation changes, studies with no follow-up time, hospitals implementing accreditation guidelines but not accredited, non-English publications, reviews, meta-analyses, master theses, and poor-quality studies were excluded.

### Results

Two authors independently applied the above criteria, following which 16 articles were analyzed. Two of these, however, were further excluded due to poor quality; 14 articles were finally included. All the articles were extremely heterogeneous, leaving no possibility for a meta-analysis.

**Data availability statement:** All relevant data are within the article and its Supporting information files.

**Funding:** This work was supported by the Orton Research Institute through grants from the Ministry of Social Affairs and Health in Finland, grant no. A2500/495.

**Competing interests:** The authors have declared that no competing interests exist.

## Conclusions

The impacts of accreditation are significant in underdeveloped than in developed nations where the legal requirements are high. Concerns regarding costs and work-load associated with accreditation processes are increasing. Moreover, studies regarding JCI accreditation and its impact on occupational health or patient safety are limited, thus warranting further investigations. PROSPERO registration number: CRD42021275665.

## 1. Introduction

Healthcare accreditations are widely used internationally to standardize medical care and hospital operations to ensure safety and efficiency. One reason is increasing medical tourism. Accreditation is a model of external evaluation that has long been used in the healthcare industry, with international accreditations gaining more positions than national ones. In the medical field, an external evaluation entails different demands and standards that hospitals must meet [1].

Accreditation aims at measuring how quality has improved over consecutive periods of time, which is why it is linked to health facility initiatives regarding quality of care. However, in practice, accreditation is often viewed from a more tangible viewpoint—that is, "accredited" or "not accredited". As a concept, accreditation differs from other health service assessments in its external, independent, and recurrent evaluation nature against quality standards as well as in having reported results that relate to recommendations and actions to improve quality at the facility's level.

Furthermore, although there is limited data on time and financial requirements associated with accreditation, it is an expensive and time-consuming process for a hospital [2]. Moreover, there is little evidence to support the positive impacts of accreditation [2], which this study focuses on, and its economic consequences are poorly understood and documented [3]. In Australia, accreditation costs 0.03%–0.6% of the total hospital operating costs per year. Mumford et al. found that smaller facilities had relatively higher costs than bigger units. This could be related to a level of fixed costs for the survey process that would be unrelated to hospital size and activity [4].

Despite the efforts and resources dedicated to the implementation of quality practices in hospital, the outcomes may fall far below expectations unless managers learn how to enable employees to perform their work effectively rather than performing them only for the sole purpose of complying with quality requirements [1]. Hence, whether accreditation has added value or not remains unclear, and more research is needed to assess its effects on the healthcare industry. In this study, we aimed to illuminate the benefits of Joint Commission International (JCI) accreditation and other experiences related to the accreditation process. We focused on the JCI accreditation because it is the most widely applied accreditation internationally. In addition, we were particularly interested in the impact of accreditation on occupational health and patient safety.

## 2. Methods

This review was registered at the International Prospective Register of Systematic Reviews: PROSPERO (CRD42021275665). We followed the Preferred Reporting Items for Systematic Reviews and Meta-Analyses (PRISMA) guidelines (S1 File) in conducting the review and reporting our findings [5].

### 2.1. Data sources and search strategy

We conducted a comprehensive literature search in December 2023 using the CINAHL, PsycINFO, ProQuest, PubMed, and Scopus databases. We searched for key terms such as "Joint Commission International" or "JCI" in the titles and abstracts of articles. No language restrictions were applied nor any date limitations. Searches were also conducted for previous systematic reviews and cross-references. Searches were repeated in September 2024 and at this point there were no new articles published that could be included in this research.

### 2.2. Inclusion and exclusion criteria

The inclusion criteria were as follows: original articles that included JCI-accredited hospitals with before–after accreditation follow-up processes, hospitals during the accreditation process, and job satisfaction and patient safety as the primary outcome measures.

The exclusion criteria were as follows: non-original articles, articles including non-accredited hospitals, studies with no follow-up time, studies with no before–after accreditation comparisons, hospitals implementing accreditation guidelines but not accredited, non-English publications, reviews, meta-analyses, and master theses.

### 2.3. Data extraction

The following data were extracted from the included articles: (a) study characteristics and design (author, year, and sample size), (b) participants, (c) intervention and comparison groups, (d) follow-up time, (e) functional tests, (f) outcome measures, (g) results, and (h) conclusions.

### 2.4. Methodological quality evaluation and risk of bias

Two independent researchers (AV and LR) extracted the data from the included articles in a standardized data collection form, and a third researcher (HH) validated the data extraction. Any disagreements between the researchers were resolved by the third researcher (HH), and consensus was attained. Quality assessments were evaluated using the Study Quality Assessment Tool criteria developed by the National Heart, Lung, and Blood Institute [6]. The risk of bias was analyzed using the Cochrane Effective Practice and Organisation of Care [7] and classified as low, high or unclear. The overall quality of the studies was assessed as good, fair, or poor (Table 1). Detailed assessments of the risk of bias and overall study quality are available in S3 and S4 Files.

## 3. Results

### 3.1. Literature screening process

Our search yielded 536 records according to the predefined search strategy, of which 247 were duplicates. All 291 records are listed in S5 File. After screening the titles, only 177 studies were included. Then, full papers of 83 articles were read to identify eligible studies. Here, the most common reasons for exclusion were missing follow-up time and only part of the JCI accreditation was implemented. Two of the authors (AV and LR) selected 16 original articles for the systematic review (Fig 1), one of which was found manually in the references.

After a detailed review, however, 2 of 16 articles were further excluded due to poor quality, lack of clearly identified before–after accreditation changes, and lack of descriptions detailing what was done during the accreditation process [11,20]. The final literature review yielded 14 articles that fulfilled the criteria (Table 2). The flow chart is presented in Fig 1.

**Table 1. Review and quality evaluation of the included studies.**

| # | Author(s) & Country code | Title | Risk of Bias | Overall Quality |
|---|---|---|---|---|
| 1 | Devkaran, S. and O'Farrell P.N. (2014) [8] (UAE) | The impact of hospital accreditation on clinical documentation compliance: a life cycle explanation using interrupted time series analysis. | Low | Good |
| 2 | Devkaran, S. and O'Farrell P.N. (2015) [9] (UAE) | The impact of hospital accreditation on quality measures: An interrupted time series analysis Quality, performance, safety and outcomes. | Low | Good |
| 3 | Devkaran, S., et al. (2019) [10] (UAE) | Impact of repeated hospital accreditation surveys on quality and reliability, an 8-year interrupted time series analysis. | Low | Good |
| 4 | Fanelli, S. et al. (2017) [11] (ITA) | The impact of regional policies on emergency department management and performance: the case of the regional government of Sicily | Unclear | Poor |
| 5 | Fang X., et al. (2016) [12] (CHN) | Safe medication management and use of narcotics in a Joint Commission International-accredited academic medical center hospital in the People's Republic of China | Low | Good |
| 6 | Halasa, Y.A., et al. (2015) [13] (JOR) | Value and impact of international hospital accreditation: a case study from Jordan. | Unclear | Fair |
| 7 | Hanumanthayya M. (2023) [14] (IND) | Nursing Practice Improvement Strategies for Reducing Medication Errors | Unclear | Fair |
| 8 | Inomata, T., et al. (2018) [15] (JPN) | The impact of Joint Commission International accreditation on time periods in the operating room: A retrospective observational study | High | Fair |
| 9 | Kagan, I., et al. (2014) [16] (ISR) | Computerization and its contribution to care quality improvement: The nurses' perspective | High | Fair |
| 10 | Kagan, I., et al. (2016) [17] (ISR) | Effect of Joint Commission International Accreditation on the Nursing Work Environment in a Tertiary Medical Center | High | Fair |
| 11 | Mekory, T.M., et al. (2016) [18] (IRN) | Evaluation of the quality of services delivered in Qazvin's Hospitals to attract medical tourists: Joint Commission International approach | High | Fair |
| 12 | Okumura, Y., et al. (2019) [19] (JPN) | Shortened cataract surgery by standardisation of the perioperative protocol according to the Joint Commission International accreditation: a retrospective observational study | Low | Good |
| 13 | Shawan, D. A. (2021) [2] SAU) | The effectiveness of the Joint Commission International accreditation in improving quality at King Fahd University Hospital, Saudi Arabia: A mixed methods approach | Low | Good |
| 14 | Song P., et al. (2014) [20] (CHN) | An outpatient antibacterial stewardship intervention during the journey to JCI accreditation | High | Good |
| 15 | Tarieh R.R.A., et al. (2021) [21] (SAU) | A case study exploring the impact of JCI standards implementation on staff productivity and motivation at the laboratory and blood bank | Unclear | Poor |
| 16 | Wang, H.F., et al. (2016) [22] (CHN) | Quality improvements in decreasing medication administration errors made by nursing staff in an academic medical center hospital: A trend analysis during the journey to Joint Commission International accreditation and in the post-accreditation era. | Low | Good |

## 3.2. Essential characteristics

Eleven (71%) of the 14 eligible studies were conducted in developing countries. The studies' key results are presented in Table 3. The most reliable of them were the ones that applied the interrupted time series (ITS) method, a quasi-experimental design (QED). The ITS design is considered the strongest among QEDs and is a powerful tool for evaluating the impact of interventions and programs implemented in healthcare settings [23]. In this review, only four studies of 14 applied ITS [2,8–10], and the rest were either before–after intervention follow-up studies or retrospective studies. The studies that used the ITS design provided the best evidence of the effects of JCI accreditation, which also explains why we had focused more on them here.

Of the four articles that used ITS, three were conducted by the same research group that considered quality and patient safety standards according to JCI. Devkaran and O´Farrell (2015) investigated performance outcomes in a 150-bed multispecialty hospital in Abu Dhabi, United Arab Emirates. The quality performance outcomes were observed either over a 48-month or 96-month period. In addition, they presented an empirical ITS analysis that was designed to examine the impact of healthcare accreditation on hospital quality measures. The quality performance differences were compared across monthly intervals between two-time segments, 1 year pre-accreditation (2009) and 3 years post-accreditation

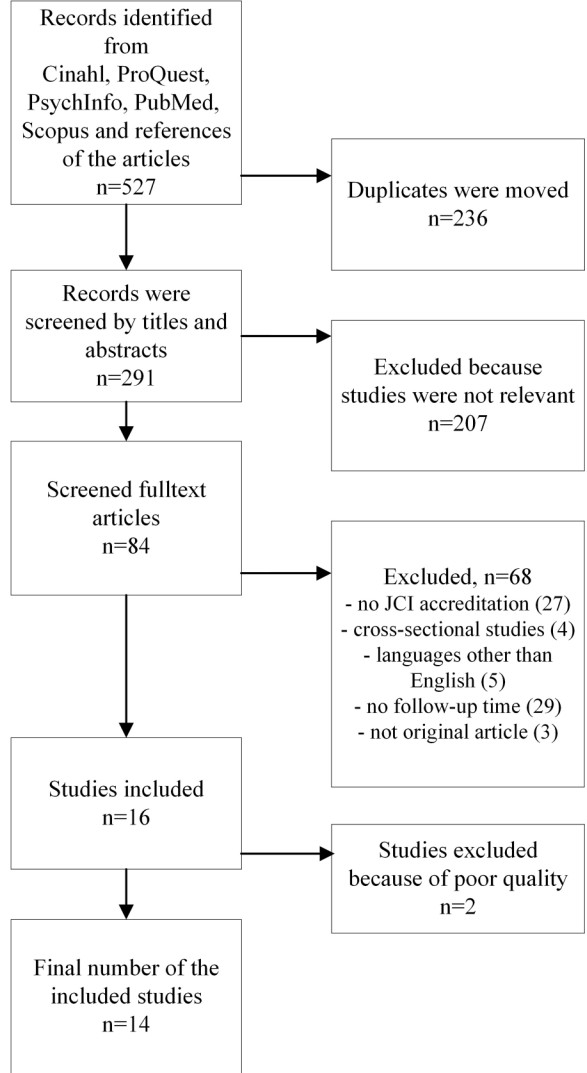

**Fig 1. Flowchart of the study selection criteria.**

(2010, 2011, and 2012) for the 27 quality measures. The principal data source was a random sample of 12,000 patient records drawn from a population of 50,000 during the study period. Each month (during the study period), a random sample of 24% of patient records was selected and audited, resulting in 324,000 observations. The measures (structure, process, and outcome) were related to dimensions of quality and patient safety. The study showed that preparation for the accreditation survey resulted in a significant improvement because 74% of the measures had a significant positive pre-accreditation slope. However, accreditation had a larger significant negative effect (48% of measures, for example percentage of completed pain reassessments and completion of the typed post-operative note within 48 hours) than a positive effect (4%, the only measure with positive impact was Turnaround time of Troponin lab results) on the post-accreditation slope of performance. Accreditation had no significant impact on 11 of the 27 measures. There was, however, residual benefit from the accreditation 3 years later, with performance maintained at approximately 90%, which was 20% higher than the baseline level in 2009 [9].

**Table 2. Study design, outcome measure, and follow-up time of the included articles.**

| Author(s) | Study design | Main Outcome | Follow-up period |
|---|---|---|---|
| Devkaran, S. and O'Farrell P.N. (2014) [8] | Interrupted time series (ITS) | Patient Safety | 48 months, 1-year pre-accreditation (2009) and 3 years post-accreditation (2010–2012). |
| Devkaran, S. and O'Farrell P.N. (2015) [9] | Interrupted time series (ITS) | Patient Safety | 48 months, 1-year pre- accreditation (2009) and 3 years post-accreditation (2010, 2011 and 2012) |
| Devkaran, S. et al. (2019) [10] | Interrupted time series (ITS) | Patient Safety | 8 years, 1-year pre accreditation (2007) and 3 years post accreditation for each of the three accreditation cycles (2008, 2011 and 2014). |
| Fang X., et al. (2016) [12] | Longitudinal follow-up study | Patient Safety | 5 years (2011–2015) |
| Hanumanthayya M. (2023) [14] | Longitudinal follow-up study | Patient Safety | 3.5 years (2015–2018) |
| Mekory, T.M., et al. (2016) [18] | Retrospective cross-sectional study | Patient Safety | 2/2014–2/2015 |
| Shawan, D. A. (2021) [2] | Interrupted time series (ITS), mixed methods | Patient Safety, Job satisfaction | 57 months (1/2014–9/2018) |
| Song P., et al. (2014) [20] | Before-after intervention study | Patient Safety | 1-year (March of 2012 – March of 2013) |
| Kagan, I., et al. (2014) [16] | Before-after intervention study | Job Satisfaction | 24 months 2008–2010 (before and after accreditation) |
| Kagan, I., et al. (2016) [17] | Before-after intervention study | Job Satisfaction | 33 months period of the JCI accreditation process (from 3 months before the JCI accreditation preparation process began to 3 months after JCI accreditation had been awarded 2.5 years later). |
| Halasa, Y.A., et al. (2015) [13] | Retrospective study | Work Effiency | 4-year period from 2006 to 2009 |
| Inomata, T., et al. (2018) [15] | Retrospective observational study | Work Effiency | 12/2014–12/2015, reaccreditation 1/2016–6/2016 |
| Okumura, Y. et al. (2019) [19] | Retrospective cross-sectional study | Work Effiency | 27 months (4/2014–6/2016) |
| Wang, H.F., et al. (2015) [22] | Intervention study | Work Effiency | 2-years (first half-year of 2012) to 64 (first half-year of 2014) |

In 2014, Devkaran and O´Farrell (2014) presented the life cycle model using ITS analysis on hospital accreditation. The developed life cycle model explained 87% of the variation in quality compliance measures [8]. In 2019, Devkaran et al. presented further results of the accreditation to evaluate whether hospital re-accreditation improves quality, patient safety, and reliability over three accreditation cycles. The validity of the life cycle model was tested by calibrating ITS regression equations for 27 quality measures. The results provided some evidence for the validity of the four phases of the life cycle: initiation phase, pre-survey phase, post-accreditation slump phase, and stagnation phase. The study showed a significant reduction in variation of the quality measures with subsequent accreditation cycles. It demonstrated that accreditation could potentially sustain improvements over the accreditation cycle, and once a high level of quality compliance has been achieved—following the first accreditation visit, it is very likely to be sustained. Repeated surveys reduced variations in quality performance, thus supporting the organization's reliability [10].

Shawan et al. (2021) utilized a convergent parallel mixed method. For the quantitative analysis, an ITS was conducted to assess the changes in a total of 12 quality outcomes pre- and post-accreditation. Thematic analysis was utilized to collect and analyze qualitative data from hospital employees and health providers [2]. The quantitative results indicated that pursuing accreditation positively impacted 9 of 12 outcomes. The improved outcomes included the average length of stay, percentage of hand hygiene compliance, rate of nosocomial infections, percentage of radiology reporting outliers, rate of pressure ulcers, percentage of the correct identification of patients, percentage of critical lab reporting, and bed occupancy rate. The outcomes that did not improve were the rate of patients leaving the emergency room without being seen, percentage of operation cancellations, and rate of patient falls. Similarly, the qualitative analysis suggested that

**Table 3. Key results of the accreditation process.**

| Author(s) | Results |
|---|---|
| Devkaran, S. and O'Farrell P.N. (2014) [8] | • Interrupted time series regression analysis of 23 quality and accreditation compliance measures were used to develop and test the Life Cycle Model on hospital accreditation<br>• The four phases of the life cycle of accreditation were recognized: the initiation phase, the presurvey phase, the post-accreditation slump phase and the stagnation phase.<br>• There was a reduction in compliance immediately after the accreditation survey, but no further subsequent fading in quality performance |
| Devkaran, S. and O'Farrell P.N. (2015) [9] | • Preparation for the accreditation survey results in significant improvement as 74% of the measures had a significant positive pre-accreditation slope<br>• Accreditation had a larger significant negative effect (48% of measures) than a positive effect (4%) on the post accreditation slope of performance<br>• There is residual benefit from accreditation three years later with performance maintained at approximately 90%, which is 20 percentage points higher than the baseline level |
| Devkaran, S. Et al. (2019) [10] | • The results provide some evidence for the validity of the four phases of the life cycle: the initiation phase, the presurvey phase, the post accreditation slump and the stagnation phase.<br>• The significant reduction in the variation of the quality measures with subsequent accreditation cycles indicating that accreditation supports the goal of high reliability. |
| Fang X., et al. (2016) [12] | • The medical oncology ward demonstrated an increase in the pain screening rate at admission from 43.5% to 100%, cancer pain control rate from 85% to 96%, and degree of satisfaction toward pain nursing from 95.4% to 100%<br>• the ratio of number of inappropriate narcotics prescriptions to total number of narcotics prescriptions for inpatients decreased |
| Halasa, Y.A., et al. (2015) [22] | • Of the 5 selected measures, 3 showed statistically significant effects (all improvements) associated with accreditation: reduction in return to intensive care unit (ICU) within 24 hours of ICU discharge; reduction in staff turnover; and completeness of medical records.<br>• The net impact of accreditation was a 1.2 percentage point reduction in patients who returned to the ICU, 12.8% reduction in annual staff turnover and 20.0% improvement in the completeness of medical records<br>• These improvements translated into total savings of US$ 593,000 in Jordan's health-care system |
| Hanumanthayya M. (2023) [13] | • The number of medication errors (ME) fell by 58.3% between the first half of 2016 and the first half of 2018<br>• During the same time, there was a 55.6% drop in high-alert drug mistakes, mostly omissions.<br>• Errors in intravenous administration also declined |
| Inomata, T., et al. (2018) [14] | • Pre-and post-accreditation procedure/ surgery time were evaluated with patients who received elective and emergency surgeries under general anesthesia<br>• The total procedure/surgery time did not change significantly. Pre-anesthesia time significantly increased, and anesthesia induction time significantly decreased<br>• The researchers conclude that quality improvement initiatives associated with time periods in the operating room can be achieved without undermining efficiency |
| Kagan, I., et al. (2014) [15] | • After the accreditation the participants ranked the role of leadership in quality improvement, the extent of their own quality control activity, and the contribution of computers to quality improvement higher than before the accreditation<br>• The higher the rating given to quality improvement leadership, the more nurses reported quality improvement activities undertaken by them and the higher nurses rated the impact of computerization on the quality of care<br>• The study demonstrated a relationship between organizational leadership and computer use by nurses for the purpose of improving clinical care. |
| Kagan et al. (2016) [16] | • Concurrent evaluation of the nursing work climate at ward level before and after accreditation<br>• Physician-nurse relations improved<br>• The involvement of social workers, dieticians, and physiotherapists increased.<br>• Support services responded more quickly to requests<br>• Management–line and staff relations became closer |
| Mekory, T.M. et al. (2016) [17] | • The number of prescribing and medication administration errors in the 2 years as preparations for the JCI accreditation process<br>• A significant reduction in prescribing errors from 6.5 to 4.2% between years 2013 and 2014 but no significant difference in administration error rates between the two periods |
| Okumura, Y. et al. (2019) [18] | • Perioperative protocol standardization shortened preprocedural, post-procedure and total procedure time in cataract surgery under local anesthesia |

*(Continued)*

**Table 3.** (Continued)

| Author(s) | Results |
|---|---|
| Shawan, D.A. et al. (2021) [2] | • Pursuing accreditation positively impacted nine out of 12 outcomes:<br>the average length of stay, the percentage of hand hygiene compliance, the rate of nosocomial infections, the percentage of radiology reporting outliers, the rate of pressure ulcers, the percentage of the correct identification of patients, the percentage of critical lab reporting, and the bed occupancy rate.<br>• The qualitative analysis suggested that the accreditation process was perceived positively by participants. Nevertheless, participants highlighted some of the drawbacks of this process: the potential bias in observation-based key performance indicators, the focus on improving process without enhancing the hospital structure, and the increased workload which can distract from patient care. |
| Song, P. Et al. (2014) [19] | • The one-year intervention program on outpatient antibacterial use during the journey to accreditation reduced the expenditure on antibacterials, improved the appropriateness of antibacterial prescriptions.<br>• The variety of antibacterials available in outpatient pharmacy decreased from 38 to 16. The proportion of antibacterial prescriptions significantly decreased (12.7% versus 9.9%).<br>• The total expenditure on antibacterials for outpatients decreased by 34.7% and the intervention program saved about 6 million Chinese Yuan Renminbi<br>(CNY) annually. |
| Wang, H.F., et al. (2015) [21] | • The medicine administration errors (MAE) were evaluated before and after accreditation<br>• The number of MAEs continuously decreased from 143 (first half-year of 2012) to 64 (first half-year of 2014), with a decrease in occurrence rate by 60.9%.<br>• The number of MAEs related to high-alert medications decreased from 32 (the second half-year of 2011) to 16 (the first half-year of 2014), with a decrease in occurrence rate by 57.9%. Omission was the top type of MAE during the first half-year of 2011 to the first half-year of 2014, with a decrease by 50% (40 cases versus 20 cases). |

the accreditation process was perceived positively by participants. However, the participants also highlighted some of the drawbacks of the process, including the potential bias in observation-based key performance indicators, the focus on improving the process without enhancing the hospital structure, and the increased workload and paperwork, which can potentially distract one from patient care [2].

The main outcomes of the studies were patient safety (8/14), work efficiency (4/14), and job satisfaction (3/14). Shawan et al. (2021) reported safety outcomes and employees' perceptions of the quality process as described above [2]. Kagan et al. (2014) investigated the association between nurses' computer use and skills, the extent of their involvement in quality control, and improvement activities on the ward and their perception of the contribution of computerization to improving nursing care. The perception of the role of leadership commitment in the success of a quality initiative was also tested. Higher ratings for quality improvement leadership directly relate to increased nurse-reported quality improvement activities and a greater perceived impact of computerization on care quality [15]. This observation indirectly indicates the role of work satisfaction in the accreditation process, which is more clearly addressed by Kagan et al. (2016). The study question was, "How would a tertiary hospital's nursing staff respond to the huge improvement effort required for external accreditation if they were encouraged to lead the change process themselves?" The results were positive—physician–nurse relations improved; the involvement of social workers, dieticians, and physiotherapists increased; support services responded more quickly to requests; and management–line staff relations became closer [16].

Halasa et al. (2015) assessed the economic impact of JCI hospital accreditation on five structural and outcome hospital performance measures in Jordan. A 4-year retrospective study compared two private accredited acute general hospitals with matched non-accredited hospitals using difference-in-difference and adjusted covariance analyses to test the impact and value of accreditation on hospital performance measures. Of the five selected measures, three showed statistically significant effects (all improvements) associated with accreditation: reduced return to intensive care unit (ICU) within 24 h of ICU discharge, reduced staff turnover, and completeness of medical records. The net impact of accreditation was a 1.2% reduction in patients who returned to the ICU, 12.8% reduction in annual staff turnover, and 20.0% improvement

in the completeness of medical records. Pooling both hospitals over 3 years, these improvements translated into a total savings of US$ 593,000.

In four studies, safety issues focused on medication errors as part of the accreditation process [13,17,19,21], and positive changes were observed in all of them—reduced antibacterial use, improved appropriateness of antibacterial prescriptions [19], reduced medication administration errors by nurses [21], reduced medication prescription errors [17], and reduced medication errors [13].

Two studies dealt with operation time. Okumura et al.'s (2019) study showed that the total procedure time for cataract surgery under local anesthesia shortened after the accreditation process [18]. In Inomata et al.'s (2018) study, the accreditation process did not affect operation time [14]. The patients received elective or emergency surgeries under general anesthesia [14,18].

Positive changes were observed following the accreditation process in all 14 studies, except in Inomata et al.'s (2018) study. The researchers noted that quality improvement initiatives associated with operating room turnaround time can be achieved without undermining efficiency [14].

## 4. Discussion

Eleven (71%) of the 14 studies were conducted in underdeveloped countries. None of them were European studies. Similarly, Devkaran et al. (2015) observed that hospital accreditation particularly favored developing countries to guarantee quality and patient safety. These issues are equally important in developed countries. However, given that government regulations are more stringent in developed than underdeveloped countries, it somewhat diminishes their interest in accreditation [9].

All these studies have shown improvement in some areas concerning patient safety following JCI accreditation. To obtain JCI accreditation, one must show improvement in some areas of hospital quality assurance. This has promoted various development projects, and here, medical errors were often chosen as a target for quality development [8, 9, 10].

Kagan studies focused on personnel, management, and job satisfaction; precisely, how personnel perceived quality improvement work, and how nurse empowerment can yield improvement in social connections among the whole hospital personnel. These studies indicated that personnel were positively inclined to quality improvement work if properly organized [16,17]. However, the specific added value of accreditation in terms of quality improvement was difficult to identify based on these original studies alone. Although improvement in various aspects of hospital quality has been shown, these studies were very heterogeneous in terms of study design, outcomes, and follow-up time. As a result, it was difficult to deduce to what extent the attained positive outcomes could be achieved without any formal quality accreditation. What is the real role of accreditation in safety measures, such as infection control, medication management, and error prevention, remain obscure. Do the accredited hospitals demonstrate better patient outcomes and fewer safety incidents compared to non-accredited ones? How do healthcare facilities adopt and comply with accreditation requirements, and whether these efforts genuinely enhance patient safety? Further, whether accreditation considers patient feedback and engagement in improving safety measures. These are some of the tasks for future research and warrant due consideration from health policy makers.

Brouwers et al. (2022) calculated the cost of preparing for and undergoing a first and second accreditation by JCI or Qualicor Europe in acute-care hospitals in Belgium. Additional investments and direct operational costs for the first accreditation cycle amounted to €879.45 per bed, and 3.8 full-time equivalents per hospital additional new staff members were recruited to coordinate and implement the trajectory. The second accreditation survey, however, costed remarkably less, with a direct operational cost and additional investment of €222.88 per bed and less investment in additional staff (1.50 full-time equivalents). Most of the costs were attributed to consulting costs and investments in infrastructure. The median total extra cost (direct operational costs and additional investments) amounted to 0.2% of the hospital's operating income for the first accreditation cycle and 0.05% for the second cycle [24].

Brouwers et al. (2022) suggested that policymakers should be aware of these significant costs as hospitals are operating using public resources, and the budget is scarce. Identifying these costs is a necessary building block to determine the cost-effectiveness of accreditation versus other quality improvement systems. Continuation of these accreditation systems and the associated costs need further study and a thorough debate. In fact, these findings are also applicable to private hospitals, which should be equally conscious about the costs [24]. However, the economic impact of accreditation may vary depending on the country, culture, and baseline situation. Halasa et al. (2015) showed remarkable savings attributed to the accreditation in Jordania [13].

JCI is the most popular healthcare accreditation worldwide. This led us to focus only on its impacts [24], which also became our limitation. However, our observations corroborated well other recent reviews on this topic. Lewis and Hinch-cliff (2023) pointed out that although accreditation is an established quality improvement intervention and despite a growing body of research, the evidence of effect remains contested. The body of research on accreditation is largely atheoretical, incapable of precisely explaining how or why hospital accreditation may influence quality improvement. As a result, the impacts of hospital accreditation remain poorly understood [25].

### Limitations of the study

One cannot ignore the effects of accreditation that were not discussed here or perhaps only superficially discussed. For instance, increased patient confidence may be a positive result of accreditation, reassuring the patients that the healthcare facility meets national or international standards. This may also attract more patients and facilitate smoother relationships with insurers. Accreditation processes often require staff training and professional development, which can improve the competency and morale of healthcare workers but also, regrettably, personnel workload. It is also possible that we have missed some relevant articles in spite of the careful search causing a further limitation for our study.

### Conclusion

This review highlighted studies with limited to various quality improvements as a result of JCI implementation, as also agreed on by personnel in some studies. As the original studies were heterogeneous, it was difficult to differentiate the specific impacts of accreditation. Criticisms related to accreditation were also observed, such as increased workload and costs associated with accreditation cycles. Hence, further research regarding the mechanisms through which hospital accreditation could enable quality improvement is warranted. In addition, there is a growing demand for alternative quality improvement systems for accreditation.

### Supporting information

**S1 File. Prisma Checklist.**
(PDF)

**S2 File. Study Quality Assessment Tools.**
(PDF)

**S3 File. Risk of Bias.**
(PDF)

**S4 File. Risk of Bias table.**
(PDF)

**S5 File. All records listed.**
(PDF)

**S6 File. Quality assessment.**

(PDF)

## Author contributions

**Supervision:** Sanna-Maria Kivivuori, Juhana Leppilahti.

**Writing – original draft:** Anni Vuohijoki.

**Writing – review & editing:** Leena Ristolainen, Heikki Hurri.

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
