## [Decision Letter · Decision Letter 0]

PONE-D-24-51441Impact of Joint Commission International accreditation on occupational health and patient safety: A systematic reviewPLOS ONE

Dear Dr. Vuohijoki,

Thank you for submitting your manuscript to PLOS ONE. After careful consideration, we feel that it has merit but does not fully meet PLOS ONE’s publication criteria as it currently stands. Therefore, we invite you to submit a revised version of the manuscript that addresses the points raised during the review process.

We look forward to receiving your revised manuscript.

Kind regards,

Naeem Mubarak, PhD

Academic Editor

PLOS ONE

**Journal Requirements:**

This work was supported by the Orton Research Institute through grants from the Ministry of Social

Affairs and Health in Finland, grant no. A2500/495.

5. As required by our policy on Data Availability, please ensure your manuscript or supplementary information includes the following: 

**Additional Editor Comments:**

The manuscript has a good deal of merit for publication after minor revisions

Reviewers' comments:

Reviewer's Responses to Questions

**Comments to the Author**

1. Is the manuscript technically sound, and do the data support the conclusions?

Reviewer #1: Partly

2. Has the statistical analysis been performed appropriately and rigorously? 

Reviewer #1: Yes

3. Have the authors made all data underlying the findings in their manuscript fully available?

Reviewer #1: Yes

4. Is the manuscript presented in an intelligible fashion and written in standard English?

Reviewer #1: Yes

5. Review Comments to the Author

**Reviewer #1:**  Thank you for the opportunity to review your manuscript entitled “Impact of Joint Commission International Accreditation on Occupational Health and Patient Safety: A Systematic Review”. The review contributes significantly by discussing how international accreditation affects important outcomes, including patient safety and occupational health (such as job satisfaction and productivity). A notable effort was made by the authors to highlight the gaps with regards to the impact of Joint Commission International (JCI) accreditation on these outcomes.

Following suggestions are offered to further enhance the quality, clarity, and coherence in the manuscript.

1. Short Title: The short title while informative does not explicitly reflect the study design. For greater clarity and precision, a revision is suggested, such as: “A Systematic Review on Joint Commission International Accreditation in Occupational Health and Patient Safety.” This concise title clearly conveys the scope of the study while ensuring that it is aligned with the systematic review design.

2. Abstract: The abstract would benefit from a clearer structure and greater coherence. Using subheadings such as Background, Methods, Results, and Conclusion is recommended. Additionally, the Methods section would include specific details regarding the study design, data sources, and inclusion/exclusion criteria.

3. Introduction:

a. Rephrasing and Redundancy: The sentence in lines 64-65 contains multiple instances of “and,” which disrupt the introductory flow. Simplifying this sentence would improve readability and coherence, particularly given its placement at the beginning of the paper.

b. Elaboration: Lines 81-82 note that smaller facilities incur higher costs, but the statement lacks sufficient explanation. Expanding on this observation with specific examples or references, such as “Mumford et al. found that smaller facilities in Australia incurred higher costs due to [specific reasons],” would enhance clarity and provide a more complete discussion.

4. Materials and Methods:

a. Title: It is suggested to consider renaming this section to “Methods” as the study does not include experiments or materials. This would better align with the study’s scope as a systematic review.

b. Language Restrictions: The statement in line 103, “No language restrictions were applied,” contradicts the exclusion of non-English publications mentioned later in lines 114-115. Clarification is necessary to ensure consistency in the description of inclusion criteria.

c. Criteria Justification: The rationale for selecting the National Heart, Lung, and Blood Institute’s quality assessment tool should be elaborated upon. For example providing information on why this specific tool was appropriate for evaluating the methodological quality of the included studies would strengthen the methodology section. Similarly, the classification criteria (good, fair, poor) should be clearly defined. Including a table summarizing the criteria before Table 1 would improve understanding and transparency.

d. Generalizability: Table 1 could be expanded to address how the included studies are representative of global impacts and their relation to targeted regions. Including regional percentages or distributions would add depth to the discussion and demonstrate the study's broader applicability.

5. Results:

a. Negative Effects: Lines 173-174 indicate that accreditation had a larger negative impact than positive but provide limited context. Expanding on this observation with examples and analysis would help clarify the implications and provide a more balanced view.

b. Rephrasing: Lines 213-215 contain a lengthy and complex sentence that could be simplified. For instance: “Higher ratings for quality improvement leadership directly relate to increased nurse-reported quality improvement activities and a greater perceived impact of computerization on care quality.” This version is more concise while retaining the original meaning.

c. Table 3 Formatting: Certain numerical and punctuation errors in Table 3 should be corrected for accuracy and consistency. Examples include “58,3%,” which should be changed to “58.3%,” and “US$ 593 000,” which should be revised to “US$ 593,000.”

6. Discussion:

a. Content Improvement: The discussion section tends to reiterate findings already presented in earlier sections. Instead, it could focus on identifying key research gaps and emphasizing how the study addresses those gaps. Highlighting the study’s implications for policy-making, future research directions, and societal benefits would significantly enhance the section.

b. Highlighting citations focusing on why patient safety is a crucial aspect in terms of cost effectiveness and improving the overall quality of life would strengthen the impact of international accreditation in healthcare services.

c. Separate Headings: Adding distinct sections for Limitations and Conclusion would improve the structure of the discussion. The Limitations section could detail the study’s shortcomings, such as the exclusion of non-English publications, while the Conclusion could emphasize the broader implications and potential applications of the findings. Highlighting how the study supports policy-making or practical improvements would underscore its relevance.

6. PLOS authors have the option to publish the peer review history of their article (what does this mean? ). If published, this will include your full peer review and any attached files.

**Do you want your identity to be public for this peer review?** For information about this choice, including consent withdrawal, please see our Privacy Policy .

Reviewer #1: No

---

## [Author Response · Author response to Decision Letter 1]

4 May 2025

Author Response to the Editor

Editor's Comment / Journal Requirements by Number:

1. We have carefully read your requirements, and we hope that we now fulfill them.

2. We have added our funding statement to both the cover letter and the manuscript, as we were unsure if both were required. Thank you for completing the online submission form on our behalf.

3. Thank you for your comment. All relevant data are included in the manuscript. We have added the following text: Line #101–102: "After December 2023, no new articles have been published that could be included in this research."

4. Line #322–323: "It is also possible that we have missed some relevant articles despite a careful search, which presents a further limitation to our study."

5. We have elaborated on our criteria and added the following text:

• Line #124: "The risk of bias was analyzed using the Cochrane Effective Practice and Organization Care (EPOC) tool and classified as low, high, or unclear."

• We have also added our quality assessment tools and risk of bias evaluations to the appendices.

6. We identified a couple of errors in our reference list and have corrected them.

Additional Editor Comment:

We are happy to hear your opinion about our article, and we sincerely hope our revision now meets your criteria.

Author response to reviewers

Reviewer 1:

Overall Response: Thank you for your clear and helpful comments.

1. Short Title: The short title while informative does not explicitly reflect the study design. For greater clarity and precision, a revision is suggested, such as: “A Systematic Review on Joint Commission International Accreditation in Occupational Health and Patient Safety.” This concise title clearly conveys the scope of the study while ensuring that it is aligned with the systematic review design.

Thank you for your comment. We have changed our short title as suggested: "A Systematic Review on Joint Commission International Accreditation in Occupational Health and Patient Safety."

2. Abstract: The abstract would benefit from a clearer structure and greater coherence. Using subheadings such as Background, Methods, Results, and Conclusion is recommended. Additionally, the Methods section would include specific details regarding the study design, data sources, and inclusion/exclusion criteria

We have clarified the introduction and restructured the abstract with subheadings: Background, Methods, Results, and Conclusion.

3. a) Rephrasing and Redundancy: The sentence in lines 64-65 contains multiple instances of “and,” which disrupt the introductory flow. Simplifying this sentence would improve readability and coherence, particularly given its placement at the beginning of the paper.

We revised the sentence to: "Healthcare accreditations are widely used internationally to standardize medical care and hospital operations to ensure safety and efficiency. One reason is increasing medical tourism."

b) Title: It is suggested to consider renaming this section to “Methods” as the study does not include experiments or materials. This would better align with the study’s scope as a systematic review.

"Examples have been added to this section. Furthermore, an incorrect reference to Mumford et al. has been corrected. Mumford et al. reported that smaller facilities incurred relatively higher costs compared to larger units. This may be attributed to fixed costs associated with the survey process, which are not dependent on hospital size or activity (4)."

4. a) Title: It is suggested to consider renaming this section to “Methods” as the study does not include experiments or materials. This would better align with the study’s scope as a systematic review.

"The headline has been modified in accordance with your suggestion."

b) Language Restrictions: The statement in line 103, “No language restrictions were applied,” contradicts the exclusion of non-English publications mentioned later in lines 114-115. Clarification is necessary to ensure consistency in the description of inclusion criteria.

"Thank you for your comment. As stated in line 103, no language restrictions were applied during the search phase. However, studies in languages other than English were excluded, as described in lines 114–115."

c) Criteria Justification: The rationale for selecting the National Heart, Lung, and

Thank you for your comment. The name of the selected assessment tool may be somewhat misleading. The NHLBI has developed a general quality assessment tool for before–after studies with no control group. This tool is not specific to heart and lung diseases. However, it was well suited to the needs of our study. While alternative tools are available, we consider this one to be as appropriate as any other.

We have added the following sentence to the Methods section:

“The NHLBI has developed a general quality assessment tool for before–after studies with no control group; therefore, the tool is not disease-specific. It was well suited to the needs of our study and was therefore selected.”

Classification Criteria

Three reviewers (AV, LR, and HH) independently read the articles and subsequently held a joint meeting, during which a consensus was reached regarding the final grading. The full evaluation system is presented in Appendix I. In addition, the risk of bias was assessed separately according to the guidelines provided at epoc.cochrane.org/resources/epoc-resources-review-authors. The corresponding guideline sheet is included in Appendix II.

Grading Scale and Criteria:

Good = The study question or objective was clearly stated. Outcome measures were prespecified, clearly defined, valid, reliable, and assessed consistently across all study participants. The study applied an interrupted time series design, and no serious sources of bias were identified.

Fair = More missing or unclear data than in studies graded as "Good."

Poor = Outcomes were not clearly prespecified, and the interrupted time series design was not applied.

d) Generalizability: Table 1 could be expanded to address how the included studies are representative of global impacts and their relation to targeted regions. Including regional percentages or distributions would add depth to the discussion and demonstrate the study's broader applicability.

Thank you for your comment.

We have added all countries to Table 1. In addition, the following sentences have been added to line 250: “Eleven (71%) of the 14 studies were conducted in underdeveloped countries. None of them were European studies.”

Furthermore, we have revised the structure of the Discussion section to better highlight the open questions and uncertainties that remain following this study.

5.

a) Negative Effects: Lines 173-174 indicate that accreditation had a larger negative impact than positive but provide limited context. Expanding on this observation with examples and analysis would help clarify the implications and provide a more balanced view.

Thank you for your comment.

We have incorporated the examples into the text as follows:

“However, accreditation had a significantly larger negative effect (affecting 48% of the measures—for example, the percentage of completed pain reassessments and the completion of the typed post-operative note within 48 hours) than a positive effect (4%; the only measure with a positive impact was the turnaround time of troponin lab results) on the post-accreditation performance slope. Accreditation had no significant impact on 11 of the 27 measures.”

b) Rephrasing: Lines 213-215 contain a lengthy and complex sentence that could be simplified. For instance: “Higher ratings for quality improvement leadership directly relate to increased nurse-reported quality improvement activities and a greater perceived impact of computerization on care quality.” This version is more concise while retaining the original meaning.

"We appreciate your clear and constructive comment. The suggested changes have been implemented accordingly."

c) Table 3 Formatting: Certain numerical and punctuation errors in Table 3 should be corrected for accuracy and consistency. Examples include “58,3%,” which should be changed to “58.3%,” and “US$ 593 000,” which should be revised to “US$ 593,000.”

We appreciate your comment. The noted errors have been addressed.

6.

a) Content Improvement: The discussion section tends to reiterate findings already presented in earlier sections. Instead, it could focus on identifying key research gaps and emphasizing how the study addresses those gaps. Highlighting the study’s implications for policy-making, future research directions, and societal benefits would significantly enhance the section.

Thank you for your comment.

We reviewed our Discussion section carefully and made substantial revisions. The following paragraph is one example of the changes made:

"The actual role of accreditation in improving safety measures—such as infection control, medication management, and error prevention—remains unclear. Do accredited hospitals demonstrate better patient outcomes and fewer safety incidents compared to non-accredited ones? How do healthcare facilities adopt and comply with accreditation requirements, and do these efforts genuinely enhance patient safety? Furthermore, does accreditation take into account patient feedback and engagement in improving safety practices? These are important questions for future research and deserve careful consideration by health policy makers."

b) Highlighting citations focusing on why patient safety is a crucial aspect in terms of cost effectiveness and improving the overall quality of life would strengthen the impact of international accreditation in healthcare services. AND

c) Separate Headings: Adding distinct sections for Limitations and Conclusion would improve the structure of the discussion. The Limitations section could detail the study’s shortcomings, such as the exclusion of non-English publications, while the Conclusion could emphasize the broader implications and potential applications of the findings. Highlighting how the study supports policy-making or practical improvements would underscore its relevance.

Thank you for your comments and suggestions.

As mentioned in Section A, we carefully revised both the Results and Discussion sections. In addition, we added new sections on Limitations and Conclusions to strengthen the manuscript. We sincerely hope that our revised approach to presenting the study, as well as the updated references, meets your expectations.

---

## [Decision Letter · Decision Letter 1]

Impact of Joint Commission International accreditation on occupational health and patient safety: A systematic review

PONE-D-24-51441R1

Dear Dr. Anni Teija Orvokki Vuohijoki,

We’re pleased to inform you that your manuscript has been judged scientifically suitable for publication and will be formally accepted for publication once it meets all outstanding technical requirements.

Kind regards,

Naeem Mubarak, PhD

Academic Editor

PLOS ONE

Additional Editor Comments (optional):

The manuscript has a good deal of merit for publication

Reviewers' comments:

Reviewer's Responses to Questions

**Comments to the Author**

1. If the authors have adequately addressed your comments raised in a previous round of review and you feel that this manuscript is now acceptable for publication, you may indicate that here to bypass the “Comments to the Author” section, enter your conflict of interest statement in the “Confidential to Editor” section, and submit your "Accept" recommendation.

Reviewer #1: All comments have been addressed

2. Is the manuscript technically sound, and do the data support the conclusions?

Reviewer #1: Yes

3. Has the statistical analysis been performed appropriately and rigorously? 

Reviewer #1: Yes

4. Have the authors made all data underlying the findings in their manuscript fully available?

Reviewer #1: Yes

5. Is the manuscript presented in an intelligible fashion and written in standard English?

Reviewer #1: Yes

6. Review Comments to the Author

Reviewer #1: (No Response)

7. PLOS authors have the option to publish the peer review history of their article (what does this mean? ). If published, this will include your full peer review and any attached files.

**Do you want your identity to be public for this peer review?** For information about this choice, including consent withdrawal, please see our Privacy Policy .

Reviewer #1: No

---

## [Editor Report · Acceptance letter]

PONE-D-24-51441R1

PLOS ONE

Dear Dr. Vuohijoki,

I'm pleased to inform you that your manuscript has been deemed suitable for publication in PLOS ONE. Congratulations! Your manuscript is now being handed over to our production team.

Kind regards,

on behalf of

Dr Naeem Mubarak

Academic Editor

PLOS ONE